

# Fractional Governing Equations of Transient Groundwater Flow in Confined Aquifers with Multi-Fractional Dimensions in Fractional Time

**M. Levent Kavvas[1], Tongbi Tu[1], Ali Ercan[1], and James Polsinelli[1]**

[1]Hydrologic Research Laboratory, Department of Civil and Environmental Engineering, University of California, Davis, CA 95616, USA.

Correspondence to: M. Levent Kavvas (mlkavvas@ucdavis.edu)

**Abstract.** Using fractional calculus, a dimensionally-consistent governing equation of transient, saturated groundwater flow in fractional time in a multi-fractional confined aquifer is developed. First, a dimensionally-consistent continuity equation for transient groundwater flow in fractional time and in a multi-fractional, multi-dimensional confined aquifer is developed. For the equation of water flux within a multi-fractional multi-dimensional confined aquifer, a dimensionally consistent equation is also developed. The governing equation of transient groundwater flow in a multi-fractional, multi-dimensional confined aquifer in fractional time is then obtained by combining the fractional continuity and water flux equations. To illustrate the capability of the proposed governing equation of groundwater flow in a confined aquifer, a numerical application of the fractional governing equation to a confined aquifer groundwater flow problem was also performed.

## 1. Introduction

Previous laboratory and field studies (Levy and Berkowitz, 2003; Silliman and Simpson, 1987; Peaudecerf and Sauty, 1978; Sidle et al., 1998; Sudicky et al., 1983) demonstrated substantial deviations from Fickian behavior in transport in subsurface porous media. Various authors (Meerschaert et al., 1999; Benson et al., 2000a, b; Schumer et al., 2001; Meerschaert et al., 2002; Baeumer et al., 2005; Baeumer and Meerschaert, 2007; Meerschaert et al., 2006; Zhang et al., 2007; Schumer et al., 2009; Zhang and Benson, 2008; Zhang et al., 2009) have introduced the fractional advection-dispersion equation (fADE) as a model for transport in heterogeneous subsurface media as one approach to the modelling of the generally non-Fickian behavior of transport. As was demonstrated by the above studies, the heavy tailed non-Fickian dispersion in subsurface media can be modelled well by a fractional spatial derivative, and the long particle waiting times in transport can be modelled well by means of a fractional time derivative within fADE. However, the above-mentioned studies focused on the fractional differential equation modeling of solute transport in fractional time-space, and not on the modeling of the underlying subsurface flows that transport the solutes. Also, as shown by Kim et al. (2014), non-Fickian behavior in transport can also be obtained if the underlying flow field has long-memory in time, which can be described by a time-fractional governing equation of the specific flow field (Ercan and Kavvas, 2014; Ercan and Kavvas, 2016). Kang et al. (2015) also showed that velocity correlation and distribution in fractured media may lead to non-Fickian transport, and proposed a Continuous Time Random Walk model (see Metzler and Klafter (2000) for details of such models) that can account for velocity correlation and distribution.

Cloot and Botha (2006) argued that there are many fractured rock aquifers where the groundwater flow does not fit conventional geometries (Black et al., 1986), and in such aquifers the conventional radial



groundwater flow model underestimates the observed drawdown in early times and overestimates it at later times (Van Tonder et al., 2001). Based on this argument, which they supported by some field radial flow data, Cloot and Botha (2006) then formulated a fractional governing equation for radial groundwater flow in integer time but fractional space and provided some numerical applications of this model. In that formulation they also

provided a formulation of the Darcy's flux in radial fractional space. However, besides taking the time as integer, they also considered a uniform homogeneous aquifer with a constant hydraulic conductivity. In the formulation of their radial groundwater flow model, they did not provide a derivation of the mass conservation equation for groundwater flow in fractional time-space. Also, they utilized the Riemann-Liouville form of the fractional derivative. Later, Atangana and his co-workers (Atangana, 2014; Atangana and Bildik, 2013;

Atangana and Vermeulen, 2014) developed the fractional radial groundwater flow formulation of Cloot and Botha (2006) in terms of the Caputo derivative, and claimed it yielded superior performance when compared to the Riemann-Liouville derivative formulation. The fundamental advantage of the Caputo derivative over the Riemann-Liouville derivative is that it can accommodate the real-life initial and boundary conditions while the Riemann-Liouville derivative cannot (Podlubny, 1998). That is, the fractional differential equations with

Caputo derivatives contain the physically-interpretable integer-order derivatives at the initial times and at the upstream spatial boundaries whereas the Riemann-Liouville derivatives do not (Podlubny, 1998). More recently, Atangana and Baleanu (2014) utilized a new definition of the fractional derivative, called the "conformable derivative" (Khalil et al. (2014)) for the modeling of radial groundwater flow in fractional time but integer space. In all the above studies, the authors formulated their fractional governing equations instead of

providing derivations of their groundwater flow equations from the basic conservation principles.

Wheatcraft and Meerschaert (2008) were the first to provide a comprehensive derivation of the continuity equation for groundwater flow. These authors have shown that since a first-order Taylor series approximation is used to represent the change in the mass flux through a control volume, the traditional continuity equation in an infinitesimal control volume is exact only when the change in flux in the control volume is linear. They also

showed that in analogy to using a first-order Taylor series, a fractional Taylor series is able to represent the nonlinear flux in a control volume exactly by only two terms. By replacing the integer-order Taylor series approximation for flux with the fractional-order Taylor series approximation, they derived a fractional form of the continuity equation for groundwater flow, removing the linearity or piecewise linearity restriction for the flux, and the restriction that the control volume must be infinitesimal. In their development of the continuity

equation, Wheatcraft and Meerschaert (2008) considered the porous medium in fractional space but the flow process in integer time. They also considered the fractional porous media space to have the same fractional power in all directions. Furthermore, their derivation is confined to only the mass conservation. It does not address the fractional water flux (motion) equation, nor the complete governing equation of groundwater flow.

Groundwater level fluctuations through time at certain locations exhibit long-range time correlation, which

implies the need for the incorporation of time-fractional operation in the standard groundwater flow governing





equations in order to accommodate the long-range time dependence (Li and Zhang, 2007; Rakhshandehroo and Amiri, 2012; Tu et al., 2017; Yu et al., 2016). Hence, in order to provide a general modeling structure, it is necessary to develop the governing equations of confined groundwater flow in fractional time as well as in fractional space. Also, different fractional powers should be considered in different spatial directions in order to accommodate the anisotropy of a confined aquifer medium.

In parallel to the conventional governing equations of groundwater flow processes (Bear, 1979; Freeze and Cherry, 1979), the corresponding time-space fractional governing equations of the confined groundwater flow must have certain characteristics (Kavvas et al. 2017): a) From the outset, the form of the governing equation must be known completely. As such, it must be a prognostic equation. That is, in order to describe the evolution of the flow field in time and space it is solved from the initial conditions and boundary conditions. The governing equation is fixed throughout the simulation time and space for the simulation of the groundwater flow in question once its physical parameters, such as porosity, saturated hydraulic conductivity, etc., are estimated. b) The fractional governing equations must be purely differential equations, containing only differential operators, and no difference operators. c) These equations must be dimensionally consistent. d) As the orders of the fractional derivatives in the equations approach the corresponding integer powers, the fractional governing equations of confined groundwater flow with fractional powers must converge to the corresponding conventional governing equations with integer powers. The following development of the fractional governing equations of confined groundwater flow will be performed within the above framework.

## 2. Derivation of the Continuity Equation for Transient Groundwater Flow in a Multi-Fractional Confined Aquifer in Fractional Time

Let $D_a^{k\beta} f(x)$ be a Caputo fractional derivative of the function f(x), defined as ( Li et al., 2009; Odibat and Shawagfeh, 2007; Podlubny, 1998; Usero, 2007),

$$D_a^{k\beta} f(x) = \frac{1}{\Gamma(m-k\beta)} \int_a^x \frac{f^m(\xi)}{(x-\xi)^{k\beta+1-m}} d\xi \qquad , \qquad m\text{-}1< \beta < m, \ m\epsilon N, \ x \geq a \quad . \tag{1}$$

Specializing the integer $m$ =1 reduces equation (1) to

$$D_a^{k\beta} f(x) = \frac{1}{\Gamma(1-k\beta)} \int_a^x \frac{f^{`}(\xi)}{(x-\xi)^{k\beta}} d\xi \ , \qquad 0 < \beta < 1, \ \ x \geq a \quad . \tag{2}$$

Then to $\beta$-order

$$D_a^{\beta} f(x) = \frac{1}{\Gamma(1-\beta)} \int_a^x \frac{f^{`}(\xi)}{(x-\xi)^{\beta}} d\xi \qquad 0 < \beta < 1, \ \ x \geq a \quad . \tag{3}$$





One can obtain a $\beta_{x_i}$-order approximation (i=1,2,3 ; $x_1 = x$ , $x_2 = y$, $x_3 = z$ ) to a function f ($\cdot$) around "a" as

$$f(x_i) = f(a) + \frac{(x_i-a)^{\beta_{x_i}}}{\Gamma(\beta_{x_i}+1)} D_a^{\beta_{x_i}} f(x_i), \; 0 < \beta_{x_i} < 1 \;\; ; \;\; i=1,2,3 \qquad x_1 = x, x_2 = y, x_3 = z \qquad (4)$$

This result may be obtained by taking in the mean value representation of a function in terms of fractional

Caputo derivative (Odibat and Shawagfeh, 2007; Usero, 2007; Li et al., 2009) the upper limit value of the

Caputo derivative at "$x_i$" (i=1,2,3; $x_1 = x$ , $x_2 = y$, $x_3 = z$) to have a distinct value for the above $\beta_{x_i}$-order

approximation (i=1,2,3 ; $x_1 = x$ , $x_2 = y$, $x_3 = z$ ) of the function f around "a". Based on this approximation,

for the whole modelling domain in time and space, the governing equations become prognostic equations that

shall be known from the outset of model simulation. The next issue is what to take for the value of "a". If one

expresses equation (4) with   a = $x_i$ - $\Delta x_i$, that is,

$$f(x_i) = f(x_i - \Delta x_i) + \frac{(\Delta x_i)^{\beta_{x_i}}}{\Gamma(\beta_{x_i}+1)} D_{x_i-\Delta x_i}^{\beta_{x_i}} f(x_i) \qquad ; i=1,2,3 \;\; ; \;\; x_1 = x, x_2 = y, x_3 = z \qquad (5)$$

then the question becomes what to take for the value of $\Delta x_i$ in Equation (5). In order to obtain fractional

governing equations as purely differential equations, an analytical relationship between $\Delta x_i$ and $(\Delta x_i)^{\beta}$

(i=1,2,3 ; $x_1 = x$ , $x_2 = y, x_3 = z$ ) that will be universally applicable throughout the modelling domain, must

be established. Such an analytical relationship is possible when the lower limit in the above Caputo derivative in

equation (5) is taken as zero (that is, $\Delta x_i = x_i$) for f($x_i$) = $x_i$. As will be shown below, it will be possible to

develop purely differential forms (with no finite difference operators) for the fractional governing equations of

confined groundwater flow by following the above construct.

The net mass flux through the control volume in Figure 1, that also has a sink/source mass flux  $q_v \Delta x \Delta y \Delta z$,

can be formulated within the above framework as

$$[\rho q_x(x,y,z;t) - \rho q_x(x-\Delta x,y,z;t)]\Delta y \Delta z + [\rho q_y(x,y,z;t) - \rho q_y(x,y-\Delta y,z;t)]\Delta x \Delta z +$$

$$[\rho q_z(x,y,z;t) - \rho q_z(x,y,z-\Delta z;t)]\Delta x \Delta y + \rho q_v \Delta x \Delta y \Delta z \qquad (6)$$

Then by combining equation (5) with equation (6) with $\Delta x_i = x_i$ (i=1,2,3 ; $x_1 = x$ , $x_2 = y, x_3 = z$ ) and

expressing the resulting Caputo derivative $D_0^{\beta_{x_i}} f(x_i)$ (taking $\Delta x_i = x_i$ causes the lower limit in the

Caputo derivative of equation (5) to become 0) by   $\frac{\partial^{\beta_{x_i}} f(x_i)}{(\partial x_i)^{\beta_{x_i}}}$ , (i=1,2,3 ; $x_1 = x$ , $x_2 = y, x_3 = z$ ) for

convenience, yields the net mass flux through the control volume in Figure 1 to the orders of $(\Delta x)^{\beta_x}$,

$(\Delta y)^{\beta_y}$, and $(\Delta z)^{\beta_z}$ as



$$\frac{1}{\Gamma(\beta_x+1)}\left(\frac{\partial}{\partial x}\right)^{\beta_x}\left(\rho q_x(x,y,z;t)\right)(\Delta x)^{\beta_x}\Delta y \Delta z +$$

$$\frac{1}{\Gamma(\beta_y+1)}\left(\frac{\partial}{\partial y}\right)^{\beta_y}\left(\rho q_y(x,y,z;t)\right)\Delta x(\Delta y)^{\beta_y}\Delta z + \qquad (7)$$

$$\frac{1}{\Gamma(\beta_z+1)}\left(\frac{\partial}{\partial z}\right)^{\beta_z}\left(\rho q_z(x,y,z;t)\right)\Delta x\Delta y(\Delta z)^{\beta_z} + \rho q_v \Delta x\Delta y\Delta z$$

where, due to the anisotropy in the hydraulic conductivities and in the subsequent flows in the porous media, different powers for fractional derivatives are considered in the three Cartesian directions in space.

From equation (5) it also follows with $f(x_i) = x_i$ that to the order of $(\Delta x_i)^{\beta_{x_i}}$, i=1,2,3,

$$\Delta x_i = \frac{(\Delta x_i)^{\beta_{x_i}}}{\Gamma(\beta_{x_i}+1)}\frac{\partial^{\beta_{x_i}} x_i}{(\partial x_i)^{\beta_{x_i}}} \quad , \qquad \text{i=1,2,3} \ ; \ x_1 = x, x_2 = y, x_3 = z \quad . \qquad (8)$$

Also for the Caputo derivative:

$$\frac{\partial^{\beta_{x_i}} x_i}{(\partial x_i)^{\beta_{x_i}}} = \frac{x_i^{1-\beta_{x_i}}}{\Gamma(2-\beta_{x_i})} \quad , \text{i=1,2,3} \qquad ; \ x_1 = x, x_2 = y, x_3 = z \qquad (9)$$

Hence, introducing equation (9) into equation (8) yields to $\beta_{x_i}$-order fractional increments in space in the i-th direction, i=1,2,3,

$$(\Delta x_i)^{\beta_{x_i}} = \frac{\Gamma(\beta_{x_i}+1)\Gamma(2-\beta_{x_i})}{x_i^{1-\beta_{x_i}}}\Delta x_i \ , \ x_1 = x, x_2 = y, x_3 = z; $$

$$\beta_{x_1} = \beta_x, \beta_{x_2} = \beta_y, \beta_{x_3} = \beta_z \quad . \qquad (10)$$

Combining equations (10) and (7) yields for the net mass outflow through the control volume in Figure 1 as (to the order of $(\Delta x_i)^{\beta_{x_i}}$, i=1,2,3; $x_1 = x, x_2 = y, x_3 = z$),

$$\frac{\Gamma(2-\beta_x)}{x^{1-\beta_x}}\left(\frac{\partial}{\partial x}\right)^{\beta_x}\left(\rho q_x(\bar{x};t)\right)\Delta x\Delta y\Delta z + \frac{\Gamma(2-\beta_y)}{y^{1-\beta_y}}\left(\frac{\partial}{\partial y}\right)^{\beta_y}\left(\rho q_y(\bar{x};t)\right)\Delta y\Delta x\Delta z$$

$$+ \frac{\Gamma(2-\beta_z)}{z^{1-\beta_z}}\left(\frac{\partial}{\partial z}\right)^{\beta_z}\left(\rho q_z(\bar{x};t)\right)\Delta z\Delta x\Delta y + \rho q_v \Delta x\Delta y\Delta z \quad , \quad \bar{x} = (x,y,z). \qquad (11)$$

Denoting the porosity, which is the water volume per volume of the control volume in Figure 1 under saturated conditions, by n , the change of mass within the control volume in Figure 1 per time increment $\Delta t$ may be expressed as (Freeze and Cherry, 1979),

$$(\rho n|_t - \rho n|_{t-\Delta t})/\Delta t \qquad (12)$$

Meanwhile, the specific storage $S_s$ of a saturated aquifer may be defined as the volume of water that is released from a unit volume of the aquifer under a unit decline in the hydraulic head h (Freeze and Cherry,

1979). Under this definition the change of mass in the control volume of Figure 1 per time increment $\Delta t$ may be expressed as (Freeze and Cherry, 1979) ,

$$\frac{(\rho n|_t - \rho n|_{t-\Delta t})}{\Delta t} = \frac{\rho S_s(h|_t - h|_{t-\Delta t})}{\Delta t}\Delta x\Delta y\Delta z = \rho S_s \frac{\Delta h}{\Delta t}\Delta x\Delta y\Delta z \qquad (13)$$

Expressing the relationship (10) to $\alpha$-order fractional increments in time;

$$(\Delta t)^\alpha = \frac{\Gamma(\alpha+1)\Gamma(2-\alpha)}{t^{1-\alpha}}\Delta t \qquad (14)$$



Meanwhile, using the approximation (5) in the time dimension to the order of $(\Delta t)^\alpha$ , for any function g of time,

$$g(t) - g(t - \Delta t) = \frac{(\Delta t)^\alpha}{\Gamma(\alpha+1)} \left(\frac{\partial}{\partial t}\right)^\alpha g(t) \tag{15}$$

Introducing Equation (15) into the right-hand-side of Equation (13) yields to order of $(\Delta t)^\alpha$ ,

$$\rho S_s \frac{1}{\Delta t} \frac{(\Delta t)^\alpha}{\Gamma(\alpha+1)} \left(\frac{\partial}{\partial t}\right)^\alpha (h) \, \Delta x \Delta y \Delta z \tag{16}$$

Then introducing Equation (14) into expression (16) yields,

$$\rho S_s \frac{\Gamma(2-\alpha)}{t^{1-\alpha}} \left(\frac{\partial}{\partial t}\right)^\alpha (h) \, \Delta x \Delta y \Delta z \tag{17}$$

as the time rate of change of mass in the control volume of size $\Delta x \Delta y \Delta z$ .

Since the net flux through the control volume is inversely related to the time rate of change of mass within the control volume of Figure 1, one may combine Equations (11) and (17) to obtain

$$\rho S_s \frac{\Gamma(2-\alpha)}{t^{1-\alpha}} \left(\frac{\partial}{\partial t}\right)^\alpha (h) = -[\frac{\Gamma(2-\beta_x)}{x^{1-\beta_x}} \left(\frac{\partial}{\partial x}\right)^{\beta_x} \left(\rho(\bar{x};t)q_x(\bar{x};t)\right) +$$
$$+ \frac{\Gamma(2-\beta_y)}{y^{1-\beta_y}} \left(\frac{\partial}{\partial y}\right)^{\beta_y} \left(\rho(\bar{x};t)q_y(\bar{x};t)\right) + \frac{\Gamma(2-\beta_z)}{z^{1-\beta_z}} \left(\frac{\partial}{\partial z}\right)^{\beta_z} \left(\rho(\bar{x};t)q_z(\bar{x};t)\right) + \rho q_v] \tag{18}$$

In the conventional case with the integer derivatives (Freeze and Cherry, 1979),

$$\rho \frac{\partial q_{x_i}}{\partial x_i} \gg q_{x_i} \frac{\partial \rho}{\partial x_i} \quad , \text{i=1,2,3}; \ x_1 = x, x_2 = y, x_3 = z \tag{19}$$

Hence, it is also expected that

$$\rho \frac{\partial^{\beta_i} q_{x_i}}{(\partial x_i)^{\beta_i}} \gg q_{x_i} \frac{\partial^{\beta_i} \rho}{(\partial x_i)^{\beta_i}} \quad , \text{i=1,2,3}; \ x_1 = x, x_2 = y, x_3 = z \ ; \ \beta_1 = \beta_x, \beta_2 = \beta_y, \beta_3 = \beta_z \tag{20}$$

Combining the inequality (20) with the Equation (18) yields

$$S_s \frac{\Gamma(2-\alpha)}{t^{1-\alpha}} \frac{\partial^\alpha h}{(\partial t)^\alpha} =$$

$$-\frac{\Gamma(2-\beta_x)}{x^{1-\beta_x}} \left(\frac{\partial}{\partial x}\right)^{\beta_x} \left(q_x(\bar{x};t)\right) - \frac{\Gamma(2-\beta_y)}{y^{1-\beta_y}} \left(\frac{\partial}{\partial y}\right)^{\beta_y} \left(q_y(\bar{x};t)\right) - \frac{\Gamma(2-\beta_z)}{z^{1-\beta_z}} \left(\frac{\partial}{\partial z}\right)^{\beta_z} \left(q_z(\bar{x};t)\right) - q_v \tag{21}$$

$$0 < \alpha, \beta_x, \beta_y, \beta_z < 1, \ \bar{x} = (x_1, x_2, x_3)$$

as the time-space fractional continuity equation of transient groundwater flow in an anisotropic confined aquifer with fractional dimensions, and in fractional time.

Performing a dimensional analysis of Equation (21), one obtains

$$\frac{1}{T} = \frac{1}{L} \frac{1}{T^{1-\alpha}} \cdot \frac{L}{T^\alpha} = \frac{1}{L^{1-\beta_x}} \frac{1}{L^{\beta_x}} \frac{L}{T} = \frac{1}{L^{1-\beta_y}} \frac{1}{L^{\beta_y}} \frac{L}{T} = \frac{1}{L^{1-\beta_z}} \frac{1}{L^{\beta_z}} \frac{L}{T} = \frac{1}{T} \tag{22}$$

where L denotes length and T denotes time. Hence, the left hand and right hand sides of the continuity Equation
(21) for transient groundwater flow in multi-fractional space and fractional time are shown to be consistent by means of Equation (22).

It was shown by Podlubny (1998) that for $n\text{-}1 < \alpha, \beta_i < n$ where n is any positive integer, as $\alpha$ and $\beta_i$ → n, the Caputo fractional derivative of a function f(y) to order $\alpha$ or $\beta_i$ (i = 1, 2, 3; $\beta_1 = \beta_x, \beta_2 = \beta_y, \beta_3 = $



$\beta_z$) becomes the conventional n-th derivative of the function f(y). Specializing the Podlubny (1998) result to n = 1, for $\alpha$ and $\beta_i \to 1$ ( i = 1, 2, 3; $\beta_1 = \beta_x, \beta_2 = \beta_y, \beta_3 = \beta_z$), reduces the continuity equation (21) to the conventional continuity equation for transient groundwater flow in a confined aquifer:

$$S_s \frac{\partial h}{\partial t} = -\frac{\partial}{\partial x}\big(q_x(\bar{x};t)\big) - \frac{\partial}{\partial y}\big(q_y(\bar{x};t)\big) - \frac{\partial}{\partial z}\big(q_z(\bar{x};t)\big) - q_v \ . \tag{23}$$

## 3. An Equation for Specific Discharge (Motion Equation) in Fractional Multi-Dimensional Confined Aquifers

A governing equation for water flux (specific discharge) $q_{x_i}$, (i = 1, 2, 3; $x_1 = x$ , $x_2 = y, x_3 = z$ ) in a saturated or unsaturated porous medium with fractional dimensions was recently developed (Kavvas et al., 2016). For the case of transient groundwater flow in an anisotropic confined aquifer with multi-fractional dimensions that equation for the specific discharge takes the form,

$$q_i(\bar{x},t) = -K_{s,x_i}(\bar{x}) \frac{\Gamma(2-\beta_i)}{x_i^{1-\beta_i}} \frac{\partial^{\beta_i} h}{(\partial x_i)^{\beta_i}} , \text{ i} = 1,2,3; \quad x_1 = x , x_2 = y, x_3 = z \tag{24}$$

where $K_{s,x_i}(\bar{x})$ denotes the saturated hydraulic conductivity in the i-th spatial direction (i=1,2,3; $x_1 = x$ , $x_2 = y, x_3 = z$). Due to the groundwater flow being in the direction of decreasing hydraulic head, the right-hand-side (RHS) of equation (24) takes a negative sign.

A dimensional analysis on equation (24) yields L/T for the units of both the left-hand-side (LHS) and the RHS of the equation, establishing its dimensional consistency.

Applying the above-mentioned result of Podlubny (1998) on the convergence of a fractional derivative to a corresponding integer derivative, for $\beta_i \to 1$ (i = 1, 2, 3; $\beta_1 = \beta_x, \beta_2 = \beta_y, \beta_3 = \beta_z$), reduces the fractional specific discharge equation (24) for groundwater flow to the conventional Darcy's equation for groundwater specific discharge:

$$q_i(\bar{x},t) = -K_{s,x_i}(\bar{x}) \frac{\partial h(\bar{x},t)}{\partial x_i} , \text{ i} = 1,2,3 ; \quad x_1 = x , x_2 = y, x_3 = z \tag{25}$$

for the case of integer spatial dimensions. As such, the fractional specific discharge equation (24) for confined groundwater flow in fractional spatial dimensions is consistent with the conventional Darcy's equation for the integer spatial dimensions.

## 4. The Complete Equation for Transient Confined Groundwater Flow in Multi-Fractional Space and Fractional Time

One can combine the specific discharge equation (24) for groundwater flow (the motion equation) in a fractional confined aquifer with the time-space fractional continuity equation (21) of groundwater flow in fractional time-space in confined aquifers to obtain,

$$S_s \frac{\Gamma(2-\alpha)}{t^{1-\alpha}} \frac{\partial^\alpha h}{(\partial t)^\alpha} = \tag{26}$$

$$\frac{\Gamma(2-\beta_x)}{x^{1-\beta_x}} \left(\frac{\partial}{\partial x}\right)^{\beta_x} \left(K_{s,x}(\bar{x}) \frac{\Gamma(2-\beta_x)}{x^{1-\beta_x}} \frac{\partial^{\beta_x} h}{(\partial x)^{\beta_x}}\right) + \frac{\Gamma(2-\beta_y)}{y^{1-\beta_y}} \left(\frac{\partial}{\partial y}\right)^{\beta_y} \left(K_{s,y}(\bar{x}) \frac{\Gamma(2-\beta_y)}{y^{1-\beta_y}} \frac{\partial^{\beta_y} h}{(\partial y)^{\beta_y}}\right)$$



$$+ \frac{\Gamma(2-\beta_z)}{z^{1-\beta_z}} \left(\frac{\partial}{\partial z}\right)^{\beta_z} \left(K_{s,z}(\bar{x}) \frac{\Gamma(2-\beta_z)}{z^{1-\beta_z}} \frac{\partial^{\beta_z} h}{(\partial z)^{\beta_z}}\right) - q_v; \quad 0 < \alpha, \beta_x, \beta_y, \beta_z < 1; \quad \bar{x} = (x_1, x_2, x_3)$$

as the time-space fractional governing equation of transient groundwater flow in a confined anisotropic aquifer with multi-fractional dimensions and in fractional time. In Equation (26) $q_v$ may be taken as the pumping rate or recharge rate.

Performing a dimensional analysis on the governing fractional Equation (26) for confined groundwater flow results in

$$\frac{1}{T} = \frac{1}{L}\frac{1}{T^{1-\alpha}} \cdot \frac{L}{T^{\alpha}} = \frac{1}{L^{1-\beta_x}}\frac{1}{L^{\beta_x}}\frac{L}{T}\frac{1}{L^{1-\beta_x}}\frac{L}{L^{\beta_x}} = \frac{1}{L^{1-\beta_y}}\frac{1}{L^{\beta_y}}\frac{L}{T}\frac{1}{L^{1-\beta_y}}\frac{L}{L^{\beta_y}} =$$
$$\frac{1}{L^{1-\beta_z}}\frac{1}{L^{\beta_z}}\frac{L}{T}\frac{1}{L^{1-\beta_z}}\frac{L}{L^{\beta_z}} = \frac{1}{T}$$

(27)

which shows that both the RHS and the LHS of the equation have the unit $\frac{1}{T}$ which verifies its dimensional consistency.

Applying the above-mentioned result of Podlubny (1998) on the convergence of a fractional derivative to a corresponding integer derivative, for $\alpha$ and $\beta_i \to 1$ (i = 1, 2, 3; $\beta_1 = \beta_x, \beta_2 = \beta_y, \beta_3 = \beta_z$), the governing equation (26) for confined groundwater flow in fractional time-space takes the form

$$S_s \frac{\partial h(\bar{x};t)}{\partial t} = \frac{\partial}{\partial x}\left(K_{s,x}(\bar{x}) \frac{\partial h(\bar{x};t)}{\partial x}\right) + \frac{\partial}{\partial y}\left(K_{s,y}(\bar{x}) \frac{\partial h(\bar{x};t)}{\partial y}\right) + \frac{\partial}{\partial z}\left(K_{s,z}(\bar{x}) \frac{\partial h(\bar{x};t)}{\partial z}\right) - q_v,$$
$$\bar{x} = (x_1, x_2, x_3)$$

(28)

which is the conventional governing equation for transient groundwater flow in an anisotropic confined aquifer (Freeze and Cherry, 1979). As such, the time-space fractional governing equation (26) of transient groundwater flow in a confined anisotropic aquifer with multi-fractional dimensions in fractional time is consistent with the conventional governing equation for transient groundwater flow in an anisotropic confined aquifer with integer derivatives.

## 5. Physical Meaning of Fractional Time Derivative and the Non-locality of the Fractional Governing Equations of Confined Transient Groundwater Flow

Let us consider the Caputo fractional time derivative of the function f(t),

$$\frac{\partial^{\alpha} f}{(\partial t)^{\alpha}} = D_0^{\alpha} f(t)$$

(29)

defined by,

$$D_0^{\alpha} f(t) = \frac{1}{\Gamma(1-\alpha)} \int_0^t \frac{f^{'}(s)}{(t-s)^{\alpha}} ds \qquad 0 < \alpha < 1, \quad t \geq 0 \ .$$

(30)

As such, each local integer derivative $f^{'}(s)$ at each time position $s$ $(0 \leq s \leq t)$ in the time interval $(0, t)$ contributes with weight $(t - s)^{-\alpha}$ to the Caputo fractional derivative of f(t) during the time interval $(0, t)$. Hence, the Caputo derivative is a nonlocal quantity, pertaining to a time interval, versus the conventional derivative of f(t), $f^{'}(t)$, which is defined for the particular time location t. Within this framework, the effect of the initial condition at the initial time location 0 is still accounted for at any time t $(0 \leq t \leq T)$ during the





whole simulation period (0,T) by means of the fractional time derivative that appears in the above governing

equation (26) of confined transient groundwater flow in fractional time. It also follows from equation (30) that

this memory effect is modulated by the value of the fractional power $\alpha$. As shown by Podlubny (1998), as

$\alpha \rightarrow 1$, the Caputo fractional time derivative of f(t), as given by equation (30), converges to the local time

5    derivative  $f^{\cdot}(t)$ at t.

From above it follows that the above fractional governing equations are nonlocal. Accordingly, they can

account for the influence of the initial and boundary conditions on the flow process more effectively than the

corresponding local-scale integer-order conventional governing equations.

**6. A numerical application of the developed fractional governing equation of confined groundwater flow**

To illustrate the capability of the proposed governing equation of groundwater flow in a confined aquifer,

a numerical application of the fractional governing equation to the physical setting of an example from Wang

and Anderson (1995) is provided as shown in Figure 2. In this example, groundwater flow is simplified to be

one-dimensional. The length of the confined aquifer is 100 $m$. The aquifer has a hydraulic transmissivity ($T$) of

0.02 m$^2$/min and a specific storage ($S$) of 0.002. The groundwater hydraulic head is initially uniform at 20 $m$.

The water level downstream suddenly drops to 10 $m$ and stays at 10 $m$. The total simulation time is 600 minutes.

Non-dimensional groundwater hydraulic heads ($H/H_0$, where $H_0$ is the initial groundwater hydraulic

head) at $x$=50 $m$ through time in the aquifer are shown in Figure 3, when fractional derivatives in space and time

are taken as $\beta_x = \alpha = 0.8, 0.9, 1.0$. As one can see from Figure 3, compared to the curve of hydraulic head

recession in time that corresponds to $\beta_x = \alpha = 1.0$ (the conventional integer derivative case), the hydraulic

head recession in time gets slower with the decrease of $\beta_x = \alpha$ from 1. The groundwater hydraulic heads in

Figure 3 clearly show heavier tails as fractional derivative orders in space and time decrease from 1.

Additionally, the smaller the fractional orders are, the heavier the tails become with the increase in time.

**7. Discussion**

From equation (28) it may be noted that the saturated hydraulic conductivity plays the role of a diffusion

coefficient in the conventional governing equation of transient groundwater flow in an anisotropic confined

aquifer in integer time-space. Meanwhile, if one were to move the fractional time to the RHS of equation (26)

one would obtain



$$S_s \frac{\partial^\alpha h}{(\partial t)^\alpha} =$$

$$\frac{\Gamma(2-\beta_x)}{x^{1-\beta_x}} \left(\frac{\partial}{\partial x}\right)^{\beta_x} \left(K_{s,x}(\bar{x}) \frac{t^{1-\alpha}}{x^{1-\beta_x}} \frac{\Gamma(2-\beta_x)}{\Gamma(2-\alpha)} \frac{\partial^{\beta_x} h}{(\partial x)^{\beta_x}}\right) +$$

$$\frac{\Gamma(2-\beta_y)}{y^{1-\beta_y}} \left(\frac{\partial}{\partial y}\right)^{\beta_y} \left(K_{s,y}(\bar{x}) \frac{t^{1-\alpha}}{y^{1-\beta_y}} \frac{\Gamma(2-\beta_y)}{\Gamma(2-\alpha)} \frac{\partial^{\beta_y} h}{(\partial y)^{\beta_y}}\right) + \qquad (31)$$

$$\frac{\Gamma(2-\beta_z)}{z^{1-\beta_z}} \left(\frac{\partial}{\partial z}\right)^{\beta_z} \left(K_{s,z}(\bar{x}) \frac{t^{1-\alpha}}{z^{1-\beta_z}} \frac{\Gamma(2-\beta_z)}{\Gamma(2-\alpha)} \frac{\partial^{\beta_z} h}{(\partial z)^{\beta_z}}\right) - q_v \frac{t^{1-\alpha}}{\Gamma(2-\alpha)} ;$$

$$0 < \alpha, \beta_x, \beta_y, \beta_z < 1; \ \bar{x} = (x_1, x_2, x_3)$$

for the governing equation of transient groundwater flow in an anisotropic confined aquifer in fractional time-space. In this form of the governing equation of transient confined groundwater flow in fractional time-space, the saturated hydraulic conductivities are augmented by fractional powers of time, $t^{1-\alpha}$, and of space, $x_i^{1-\beta_{x_i}}$, i= 1,2,3, in terms of the ratios of fractional time to fractional space, $\frac{t^{1-\alpha}}{x_i^{1-\beta_{x_i}}}$, i= 1,2,3, in

multiple dimensions. As such the confined groundwater diffusion in fractional time-space is modulated by the above ratios of fractional time to fractional space. One can also see from the Figure 3 on the numerical application of the fractional confined groundwater flow equation to a simple one-dimensional case, as the fractional powers of the derivatives in space and time in the governing equation decrease from unity, the recession rate of the nondimensional hydraulic heads from the initial condition also gets slower with respect to

the case of the conventional governing equation with integer derivative powers.

Kavvas et al. (2014) argued and Kim et al. (2014) have shown by numerical simulations that non-Fickian behavior in solute transport can also be obtained if the underlying flow field has long-memory, which can be described by a fractional governing equation of the specific flow field. Ercan and Kavvas (2014) and Ercan and Kavvas (2016) have shown by numerical simulations that it is possible to obtain long waves in time and in

space by means of the fractional governing equations of unsteady open channel flow.

**8. Conclusion**

In this study, a dimensionally-consistent continuity equation for transient groundwater flow in multi-fractional, multi-dimensional confined aquifers in fractional time was developed. It was then shown that as the fractional powers of time and space derivatives approach unity, the time-space fractional continuity

equation approaches the conventional continuity equation for transient groundwater flow in a confined aquifer. For the motion equation of confined groundwater flow, or the equation of water flux within a multi-fractional multi-dimensional confined aquifer, a dimensionally consistent equation was also developed. It was shown that as the fractional powers of the spatial derivatives approach unity, the fractional water flux equation approaches the conventional Darcy's equation for groundwater specific discharge.

The governing equation of transient groundwater flow in multi-fractional, multi-dimensional confined aquifers and in fractional time was then obtained by combining the fractional continuity and water flux equations. It was then shown that as the fractional powers of time and space derivatives approach unity, the time-space fractional governing equation of transient confined groundwater flow approaches the conventional governing equation with integer derivatives for transient groundwater flow in an anisotropic confined aquifer.

To illustrate the capability of the proposed governing equation of groundwater flow in a confined aquifer, a numerical application of the fractional governing equation to a confined aquifer groundwater flow problem was also performed.



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



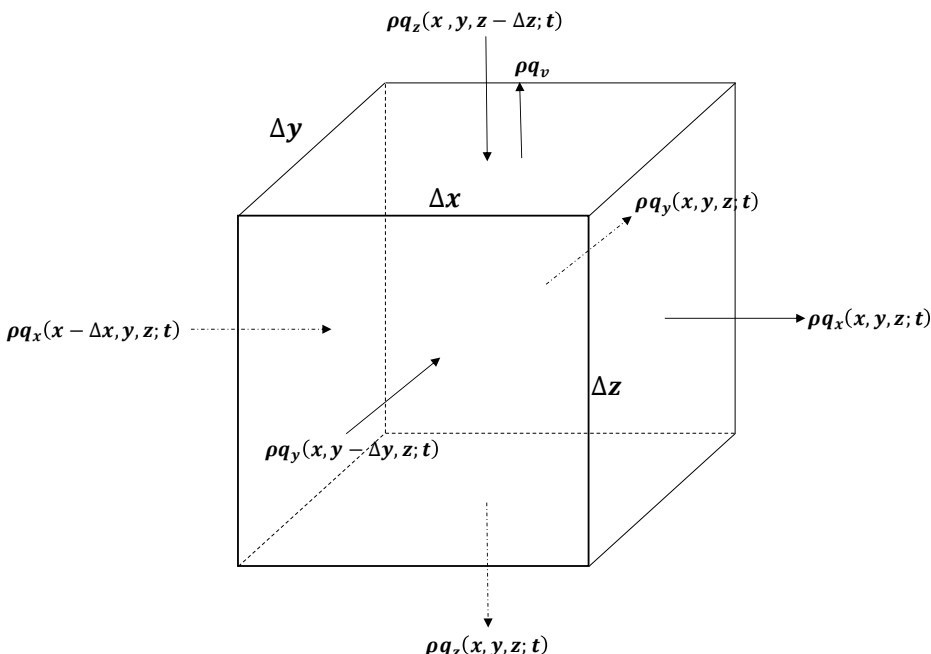

Figure 1. **The control volume for the three-dimensional groundwater flow in confined aquifers.**



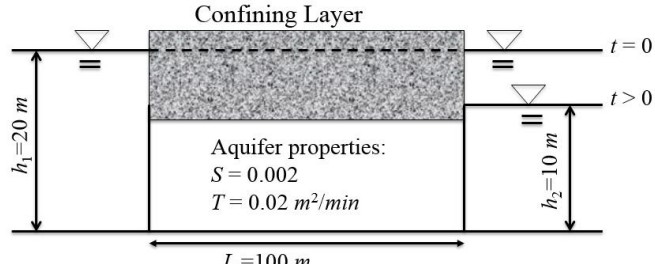

**Figure 2. The reservoir example modified based on Wang and Anderson (1995)**

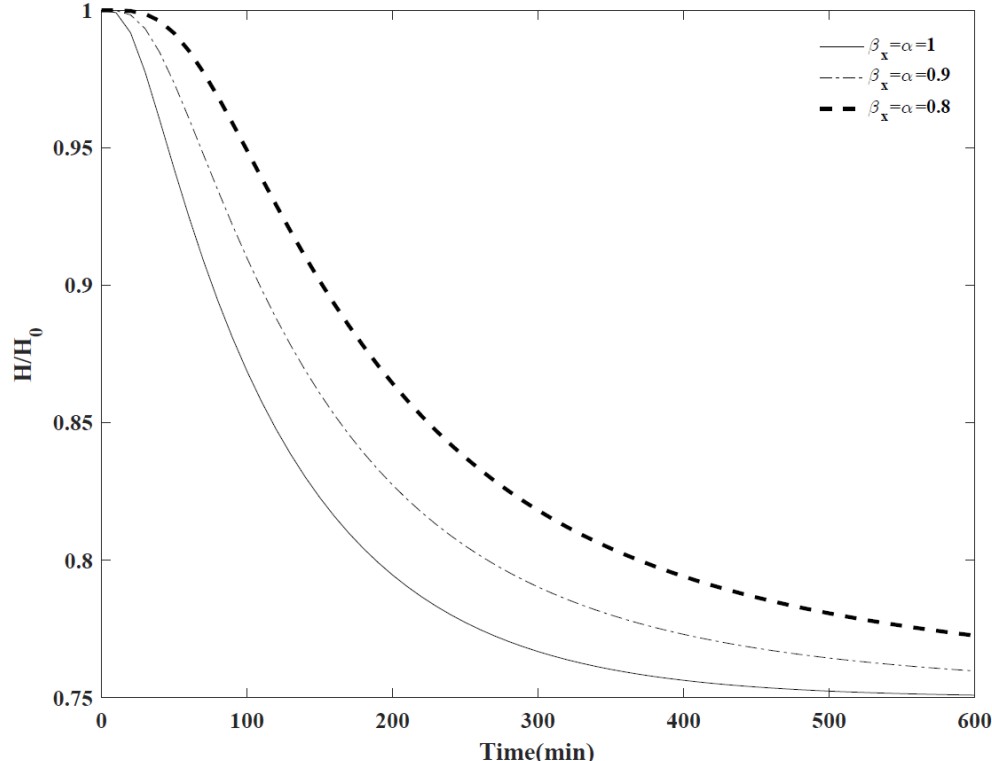

**Figure 3. Non-dimensional groundwater hydraulic heads through time at $x = L/2$ when fractional space and time derivatives are $\beta_x = \alpha = 0.8, 0.9, 1.0$, where $L$ is the length of the aquifer, $\beta_x$ and $\alpha$ are the fractional orders in space and time respectively.**

