# Peer review of "Fractional Governing Equations of Transient Groundwater Flow in Confined Aquifers with Multi-Fractional Dimensions in Fractional Time"

_Earth System Dynamics, 2017_

## Referee Comment (RC1) · Anonymous Referee #1 · 12 Jul 2017

This study developed a governing equation of transient groundwater flow in a multi-fractional, multi-dimensional confined aquifer in fractional time. This study has sufficient novelty, and the developed governing equations would be important for groundwater modeling. Furthermore, this paper is documented well. However, the application and discussion parts of this paper are short. If the authors can provide more detailed information on the application, and more discussions on the results, it would be helpful for readers to understand importance of this study.

Specific comments: 1. Mathematical symbols such as time "t" and function "f" should

be written in italic even in sentences. 2. It would be helpful to readers if the authors provide more explanation why nonlocal governing equations can account for the influence of the initial and boundary conditions on the flow process more efficiently than the corresponding local-scale equations, in P.9 L.4-6. 3. I highly recommend the authors to provide more detailed information of the setting of the numerical application in Section 6. It is possible to understand the numerical application setting if one reads Wang and Anderson (1995). However, it is currently difficult to know it from the description of this paper. 4. In P.10 L.8-10, "the recession rate of the nondimensional hydraulic heads from the initial condition also gets slower with respect to the case of the conventional governing equation with integer derivative powers." Please explain why this result is important on groundwater modeling. 5. I recommend the authors to add a description about the importance of the result of the numerical application at the end of Conclusion.

---

## Referee Comment (RC2) · Anonymous Referee #2 · 24 Jul 2017

The manuscript introduces, develops, analyses and discusses fundamental developments on fractal dynamics in both methodological and geophysical terms. Unlike the traditional fractal-geometric studies or statistical studies on scaling with non-physical considerations that abound in the literature, the present study actually provides governing equations for a highly relevant problem (transient groundwater flow in confined aquifers).

The mathematical formulations are carefully derived, explained and implemented, with a profound physical background revealing a deep understanding of the underlying sys-

tem.

For the knowledgeable reader with proper theoretical background, the manuscript will be very clear and instructive. And this is what fundamentally matters, since ESD is a scientific journal rather than an outreach magazine.

Some concepts and formulations may, however, be out of reach for soft-science i.e. purely statistical communities. This is because the authors work with proper Mathematics and produce proper Physics to advance the Geophysical Sciences in general and the Hydrological Sciences in particular.

Overall, from the eyes of a hard-line theoretician, I recognize that this is a remarkable piece of work. Congratulations.

Despite the well-deserved praise, a couple of minor remarks are now due:

a) I recommend further elaboration on the implementation and discussion of the theoretical advances introduced in the manuscript - particularly in the context of the broader Geo Sciences. That way, the interdisciplinary readership of the journal can better appreciate the developments and findings.

b) Similarly to what had been noted by the other reviewer, scalar variables or functions should be typeset in italic even in the main body of the text.

---

## Author Comment (AC1) · 31 Jul 2017

RESPONSE TO THE COMMENTS OF ANONYMOUS REFEREE #1

The authors thank the anonymous Referee #1 for the valuable comments.

Referee #1 General Comment:

"This study developed a governing equation of transient groundwater flow in a multi-fractional, multi-dimensional confined aquifer in fractional time. This study has suffi-

cient novelty, and the developed governing equations would be important for ground-water modeling. Furthermore, this paper is documented well. However, the application and discussion parts of this paper are short. If the authors can provide more detailed information on the application, and more discussions on the results, it would be helpful for readers to understand importance of this study. "

Response: The authors thank the positive comments of the Referee #1.

Referee #1 Specific comments:

1. "Mathematical symbols such as time "$t$" and function "$f$" should be written in italic even in sentences. "

Response: These changes will be made in the revised manuscript as suggested.

2. "It would be helpful to readers if the authors provide more explanation why nonlocal governing equations can account for the influence of the initial and boundary conditions on the flow process more efficiently than the corresponding local-scale equations, in P.9 L.4-6. "

Response: In P.8 L.24-P.9 L.5, the authors explained the reason why Caputo derivative can better quantify the effect of the initial and boundary conditions than the conventional derivative. As discussed in P.8 L.24, the Caputo derivative is a nonlocal quantity. The fractional governing equations based on Caputo derivative are nonlocal governing equations that can more efficiently account for the influence of the initial and boundary conditions on the flow process more efficiently than the corresponding local-scale equations. Please see also the discussion on the physical framework of fractional governing equations of soil water flow in Kavvas et al. 2016. Following the recommendation of Referee #1, more explanation will be added in the revised manuscript.

3. "I highly recommend the authors to provide more detailed information of the setting of the numerical application in Section 6. It is possible to understand the numerical application setting if one reads Wang and Anderson (1995). However, it is currently

difficult to know it from the description of this paper."

Response: The numerical example is based on the example provided in Wang and Anderson (1995). Detailed information of the numerical example used in this study was provided in P.9 L.13-16 and Figure 2: "In this example, groundwater flow is simplified to be one-dimensional. The length of the confined aquifer is 100 m. The aquifer has a hydraulic transmissivity (T) of 0.02 m2/min and a specific storage (S) of 0.002. The groundwater hydraulic head is initially uniform at 20 m. The water level downstream suddenly drops to 10 m and stays at 10 m. The total simulation time is 600 minutes." Given these information, we then solve the fractional governing equations when the orders of the fractional derivatives are 1, 0.9 and 0.8. Following the recommendation of Referee #1, more explanation will be added in the revised manuscript.

4. "In P.10 L.8-10, "the recession rate of the non-dimensional hydraulic heads from the initial condition also gets slower with respect to the case of the conventional governing equation with integer derivative powers." Please explain why this result is important on groundwater modeling."

Response: The modeling results may help explain the long-range dependence characteristics in some groundwater level datasets. The results may also shed light on the non-Fickian transport phenomena in groundwater flow. Explanation/implication of the modeling results will be added in the revised manuscript. A numerical solution methodology and additional numerical examples by the proposed fractional governing equations will be provided in Tu et al. (2017).

5. "I recommend the authors to add a description about the importance of the result of the numerical application at the end of Conclusion."

Response: Following the recommendation of Referee #1, a description about the importance/implication of the numerical application results will be added at the end of the conclusion section in the revised manuscript.
References:

Kavvas, M. L., Ercan, A., and Polsinelli, J.: Governing equations of transient soil water flow and soil water flux in multi-dimensional fractional anisotropic media and fractional time, Hydrol. Earth Syst. Sci., 21, 1547-1557, https://doi.org/10.5194/hess-21-1547-2017, 2017.

Tu, T., Ercan, A., and Kavvas, M. L.: Time-Space Fractional Governing Equations of Transient Groundwater Flow in Confined Aquifers: Numerical Investigation, Hydrological Processes,. under review, 2017.

Wang, H. F. and Anderson, M. P.: Introduction to groundwater modeling: finite difference and finite element methods, Academic Press, 1995.

RESPONSE TO THE COMMENTS OF ANONYMOUS REFEREE #2

The authors appreciate the valuable general comments of anonymous Referee #2, and thank him/her for these comments. The authors' responses to the specific comments of the referee are provided below:

" a) I recommend further elaboration on the implementation and discussion of the theoretical advances introduced in the manuscript - particularly in the context of the broader Geo Sciences. That way, the interdisciplinary readership of the journal can better appreciate the developments and findings."

Response: Following the recommendation of Referee #2, the authors will add material to the revised manuscript that will further attempt to elaborate and discuss the theoretical work in the paper in the context of broader Geosciences.

" b) Similarly to what had been noted by the other reviewer, scalar variables or functions should be typeset in italic even in the main body of the text."

Response: Following the recommendation of Referee #2, the scalar variables or functions will be typeset in italic in the revised manuscript.

---

## Author Response (AR1)

**RESPONSE TO THE EDITOR'S COMMENTS**

We thank the Editor for his positive and constructive comments. Following his recommendation, we addressed the issues that were raised in the two reviewers' reports, and revised our paper accordingly.

In the submission of the revised manuscript, we have provided a marked-up version which shows all the performed revisions.

[revised manuscript text omitted]

**6. A numerical application of the developed fractional governing equation of confined groundwater flow**

To illustrate the capability of the proposed governing equation of groundwater flow in a confined aquifer, a numerical application of the fractional governing equation to the physical setting of an example from Wang and Anderson (1995) is provided as shown in Figure 2. In this example, groundwater flow in a confined aquifer is simplified to be one-dimensional. The length of the confined aquifer is 100 $m$. The hydraulic transmissivity ($T$) of the aquifer is 0.02 m$^2$/minute and the specific storage ($S$) of the aquifer is 0.002. ute and a specific storage ($S$) of 0.002. The groundwater hydraulic head is initially uniform at 20 $m$. The water level downstream suddenly drops to 10 $m$ and stays at 10 $m$. The

groundwater level upstream is set to be 20 $m$ throughout the simulation duration. The total simulation time is 600 minutes.

Non-dimensional groundwater hydraulic heads ($H/H_0$, where $H_0$ is the initial groundwater hydraulic head) at $x$=50 $m$ through time in the aquifer are shown in Figure 3, when fractional derivatives in space and time are taken as $\beta_x = \alpha = 0.8, 0.9, 1.0$. As one can see from Figure 3, compared to the curve of hydraulic head recession in time that corresponds to $\beta_x = \alpha = 1.0$ (the conventional integer derivative case), the hydraulic head recession in time gets slower with the decrease of $\beta_x = \alpha$ from 1. The groundwater hydraulic heads in Figure 3 clearly show heavier tails as fractional derivative orders in space and time decrease from 1. Additionally, the smaller the fractional orders are, the heavier the tails become with the increase in time. The modelling results may indicate nonlocal effects in groundwater flow and help explain the long-range dependence characteristics in some groundwater level fluctuation datasets (Tu et al., 2017). The results may also shed light on the non-Fickian transport phenomena in groundwater flow.

**7. Discussion on the Developed Fractional Governing Equations in the Context of Broader Geosciences**

The conventional governing equations of porous media flows in geosciences in various environments are all local-scale equations where only the interactions among nearest neighbours in time and space are described. All of these governing equations are differential equations where the powers of the derivative terms that appear in these equations take integer values. In the case that a porous media flow field shows interactions among time-space locations that are separated by substantial distances in time or space, the local-scale conventional governing flow equations for such media, because they are based on local interactions, may not be able to describe such long-distance interactions adequately. A more efficient approach for modeling such long-distance interactions in time and space may be the use of fractional governing equations of porous media flows. Such fractional governing equations, as those developed in this study, utilize time-space derivatives with fractional powers. As already shown in Section 5 above, the fractional Caputo time derivative is nonlocal, and, as such, can accommodate the effect of the initial conditions on the groundwater flow process for times that are substantially later than the initial time. Similarly, the fractional Caputo space derivatives in the governing Equations (21), (24) and (26) of this study are also nonlocal derivatives. To see this consider the Caputo fractional space derivative $D_0^\beta f(x_i)$:

$$D_0^\beta f(x_i) = \frac{1}{\Gamma(1-\beta)} \int_0^{x_i} \frac{f'(\xi)}{(x_i-\xi)^\beta} d\xi \qquad (31)$$

Hence, each local integer derivative $f'(\xi)$ at each spatial location $\xi$ in the spatial interval $(0, x_i)$ will contribute to the Caputo fractional derivative of the interval $(0, x_i)$ with weight $(x_i-\xi)^{-\beta}$. As such, for groundwater flow in any i-direction, the effect of a boundary condition that is placed at boundary location "0" in the i-direction will be accounted for at any distance $x_i$ from the boundary location "0" by means of the fractional space derivative that appears in the above fractional governing equations for the i-th direction. It follows from Equation (31) that this effect will be modulated by the value of the fractional derivative power $\beta$ due to the weight $(x_i-\xi)^{-\beta}$.

As shown in the previous sections, the fractional governing equations converge to their conventional integer counterparts as the fractional derivative powers take integer values. Consequently, the conventional governing equations of porous media flows may be considered as special cases of the corresponding fractional governing equations, corresponding to the integer values of the derivative powers. While the fractional powers of the derivatives in the governing equation (26) may take any fractional value within the interval (0,1), the integer powers of the derivatives in the conventional governing equation (28) are restricted to the value of unity. Within this context, the fractional governing equations of porous media flows may be thought as the generalizations of the conventional governing equations of porous media flows with integer powers.

From above it follows that the fractional governing equations developed in this study are nonlocal. Accordingly, they can account for the influence of the initial and boundary conditions on the flow process more effectively than the corresponding local-scale integer-order conventional governing equations, since the conventional governing equations consider the effect of initial and boundary conditions on the flow processes within shorter time/space ranges.

[revised manuscript text omitted]